# Continuous Risk Assessment of Late and Term Preeclampsia Throughout Pregnancy: A Retrospective Cohort Study

**DOI:** 10.3390/medicina60121909

**Published:** 2024-11-21

**Authors:** Valeria Rolle, Petya Chaveeva, Ander Diaz-Navarro, Irene Fernández-Buhigas, Diana Cuenca-Gómez, Tanya Tilkova, Belén Santacruz, Teresa Pérez, Maria M. Gil

**Affiliations:** 1Faculty of Statistical Studies, Complutense University of Madrid, 28040 Madrid, Spain; 2Dr. Shterev Hospital, 1330 Sofia, Bulgaria; 3Department of Obstetrics and Gynecology, Medical University of Pleven, 5800 Pleven, Bulgaria; 4Ontario Institute for Cancer Research, Toronto, ON M5G 0A3, Canada; 5Department of Molecular Genetics, University of Toronto, Toronto, ON M5S 1A1, Canada; 6Obstetrics Department, Torrejón University Hospital, 28850 Madrid, Spain; 7Institute of Statistics and Data Science, Complutense University of Madrid, 28040 Madrid, Spain; 8School of Medicine, Faculty of Health Sciences, Francisco de Vitoria University, 28223 Madrid, Spain

**Keywords:** preeclampsia, preterm preeclampsia, screening, PlGF, sFlt-1, UtA-PI, MAP, biomarkers

## Abstract

Preeclampsia is a serious pregnancy complication affecting between 2 to 5% of pregnancies, significantly increasing health risks for both mother and baby. Traditionally, biomarkers measured in the first trimester are used to predict the risk of preeclampsia, performing well in detecting preeclampsia happening early in pregnancy, but only identify around 40–50% of cases that develop later. This study evaluates whether incorporating repeated biomarker measurements throughout pregnancy improves the detection of preeclampsia. Using data from almost 5000 pregnancies, we developed a model that integrates these routinely measured biomarkers taking all the measures into account at the same time. The addition of repeated measures improved prediction accuracy from 50% at the start of pregnancy to approximately 84% by the end. This allows for a better, more personalized approach to monitoring each pregnancy.

## 1. Introduction

Preeclampsia (PE) is a multisystemic disorder that occurs in 2–5% of pregnancies [1]. It makes a significant contribution to maternal mortality rates, and it is responsible for about 42,000 maternal deaths worldwide annually [1]. PE also increases the risk of maternal morbidity and significantly impacts neonatal health, presenting worse outcomes the earlier the onset and more severe forms of the disease [2,3].

Various risk factors among maternal and pregnancy characteristics, together with elements from the medical and obstetric history, have been identified that may predispose pregnant women to placental dysfunction [4,5,6]. Combining maternal risk factors with the first-trimester measurement of the mean arterial pressure (MAP), uterine artery pulsatility index (UtA-PI) by ultrasound, and placental growth factor (PlGF) facilitates the identification of individuals at high risk of developing PE as early as the in the first trimester [7,8,9]. The competing risk model developed by the Fetal Medicine Foundation for the first trimester has demonstrated that about 75% of women who will subsequently develop preterm PE (requiring delivery before 37 weeks) can be detected during routine first-trimester ultrasound screening at a 10% screen-positive rate (SPR). On the other hand, term PE, although generally associated with less severe outcomes, accounts for two-thirds of the PE cases, and it is, overall, the one responsible for the majority of the maternal and fetal morbidity and mortality related to this condition [10]. The prediction of term PE in the first trimester is poor, with detection rates (DR) ranging from 40 to 47% at a 10% false positive rate [7,11,12], and screening conducted in the second trimester does not improve those figures [13,14]. Conversely, third-trimester strategies may increase the DR to 53–84% [15,16,17,18]. It has therefore been proposed that readjustment of the individual risk established in the first trimester according to longitudinal changes in biomarkers obtained throughout pregnancy may improve the DR of term PE [14].

The aim of this study was to evaluate the performance of repeated measurements of biomarkers throughout pregnancy for the prediction of all (PE delivering at ≥35 weeks’ gestation) and term PE (PE delivering at ≥37 weeks’ gestation), comparing the predictive performance of the model throughout time.

## 2. Materials and Methods

### 2.1. Study Design and Population

This a longitudinal study of retrospectively extracted data, including a cohort of singleton pregnancies attending their routine ultrasound examination and first-trimester screening for preterm PE at 11 + 0 to 13 + 6 weeks’ gestation at Hospital Universitario de Torrejón (Madrid, Spain) between April 2021 and December 2022 and Shterev Hospital (Sofia, Bulgaria), between August 2017 and February 2022. All pregnant women ≥ 18 years old, with singleton pregnancies and non-malformed live fetuses were invited to participate in a larger study for the prediction and prevention of pregnancy complications. For this study, we included only women undergoing first-trimester assessment of PE, known perinatal outcome, and at least one more ultrasound assessment performed during the second and/or third trimesters of their pregnancy, where MAP and UtA-PI had been evaluated.

The study was approved by the local Research Ethics Committee at each participating hospital and written informed consent was obtained from each participant.

### 2.2. Study Procedures

During the 11 + 0 to 13 + 6 weeks hospital visit, patient characteristics and medical history were recorded in a clinical database (ViewPoint^®^ software version 5, GE Healthcare; Munich, Germany in Spain and Astraia^®^ software version 29.0.0 (DB18376), NEXUS/ASTRAIA GmbH; Ismaning, Germany in Bulgaria) including maternal age, race (White, Black, South Asian, East Asian, or Mixed), method of conception (natural or using assisted reproductive technology defined as in vitro fertilization (IVF) or use of ovulation drugs), smoking during pregnancy, weight, height, Body Mass Index (BMI, calculated as kg/m^2^), and medical and obstetric history. The obstetric history included parity (parous or nulliparous if no previous pregnancies at ≥24 weeks of gestation), and for parous women, previous PE, and birthweight and gestational age at delivery of previous baby. In this visit, we also recorded fetal crown–rump length, which was used for pregnancy dating in non-IVF pregnancies [19], MAP, mean UtA-PI and PlGF raw values. Data from all subsequent appointments performed between 18 + 0 to 37 + 0 for any reason were also retrieved from the clinical database and merged with the first-trimester data.

### 2.3. Clinical Definitions and Outcomes

Participants were followed up according to the clinical protocols of each center, and perinatal outcomes were recorded by reviewing hospital/regional records or contacting delivering hospitals or the women’s general medical practitioners/midwives.

The primary outcome was the development of PE based on the diagnostic criteria of the American College of Obstetricians and Gynecologists: (a) systolic blood pressure of 140 mm Hg or higher and/or diastolic blood pressure of 90 mm Hg or higher occurring on two occasions at least 4 h apart after 20 weeks of gestation in a woman whose blood pressure was previously normal; or (b) systolic blood pressure greater than or equal to 160 mm Hg and/or diastolic blood pressure of 110 mm Hg or higher confirmed within a short interval (minutes) and at least one of: (1) proteinuria greater than 300 mg in 24-h urine sample, protein/creatinine ratio greater than 0.3, or dipstick reading of 1+; (2) thrombocytopenia with platelet count lower than 100,000/mL; renal insufficiency with serum creatinine concentration greater than 1.1 mg/dL or a doubling of the serum creatinine concentration in the absence of other renal diseases; (3) impaired liver function suspected by elevated blood levels of liver enzymes to twice normal concentrations and pulmonary edema or cerebral or visual symptoms [20].

### 2.4. Statistical Analysis

#### 2.4.1. Descriptive Statistics

Descriptive statistics were expressed as means and standard deviations (SD), and absolute and relative frequencies, categorized by center. Comparisons between centers were conducted using the two-proportion z-test for categorical variables and the Mann–Whitney test for continuous variables.

#### 2.4.2. Model Development

The data from Sofia were used to develop the predictive model (training dataset) as the training and the data from Madrid to perform an external validation (validation dataset). This division was performed to enhance the comparability of the results as the data from Madrid come from the same population as the data presented in other validation studies [12,21]. We evaluated all biomarkers available in each assessment which included maternal characteristics and history, PlGF, MAP, UtA-PI obtained during the first trimester, and maternal weight, MAP, and UtA-PI measured in the second and third trimesters. We opted to use all available data, since standard calculations for sample size and power are not directly applicable to JM due to the dependency of the observations.

For the training dataset, all pregnancies with first-trimester assessment and at least another measurement of biomarkers throughout pregnancy were included. For the validation dataset, only women receiving first-trimester assessment and at least another measurement of UtA-PI and MAP at 35–37 weeks were included.

We fitted a joint model (JM) combining time-to-event outcome and longitudinal data to develop a model that uses all information available, using the training dataset. This approach considers that the repeated measurements used are endogenous (intrinsic to the patient) variables and measured with error, while most survival models that can accommodate time-dependent covariates assume they are exogenous and measured without error [22,23]. Independent variables included in the model were maternal age, conception (Spontaneous/Assisted), diabetes mellitus (No/Yes), previous PE (Multiparous without PE/Nulliparous/Multiparous with PE), first-trimester PlGF, BMI, aspirin intake, chronic hypertension, and the interaction of chronic hypertension and aspirin intake. All variables were retained in the model regardless of statistical significance because of their previously proven predictive value. All continuous variables were centered and scaled to help convergence and PlGF, UtA-PI, maternal age, and BMI were also transformed into the logarithmic scale, therefore the Hazard Ratios (HR) are not directly interpretable for these variables.

These MAP and UtA-PI values were modeled as longitudinal processes, using all available measurements for each pregnancy. The JM was fitted considering different alternatives for the time-to-event model; in particular, UtA-PI was modeled using its area as the functional form, meaning that instead of using the raw value of a given marker at a single time point to estimate risk, the area under the trajectory of that marker throughout the pregnancy was used. The Bayesian approach was applied to combine the time-to-event model and the longitudinal model using Markov Chain Monte Carlo (MCMC) methods. Non-informative prior distributions were considered to not influence the results and to have a fairer comparison with the first-trimester model. We used three chains initialized from different values, each chain running for 40,000 iterations with a burn-in of 500 iterations and thinning of 1. Adjusted HR (aHR) and 95% credible intervals (CdI) are reported for the survival process of the JM. This model allows for the calculation of the risk every time a woman gets an assessment, resulting in a continuous personalized prediction using all the available information. A diagnostic of this model was performed for both the longitudinal and the time-to-event models. Multicollinearity was assessed for all variables included in the model.

### 2.5. Model Validation

Using the validation dataset, we first calculated the AUROC to assess the model’s performance for term PE and all PE. The DR was also calculated at three SPR, 10%, 15%, and 20%. Next, we generated the calibration plot: the risk was categorized in bins and the observed percentage of PE was plotted against the expected; in the perfect predictions the slope would be 1 and the intercept 0. Finally, the AUROC was calculated several times throughout the pregnancy to assess if, and how much, this use of all available measurements of biomarkers improves the regular first-trimester risk assessment.

### 2.6. Software

R software in its version 4.4.1 was used for all analyses [24]. The JMbayes2 [25] and Survival [26] packages were used to fit the JM and to calculate predictions of the model. We also used table1 [27] for the tables, pROC [28] for the AUROC curves, ggplot2 [29] for the figures, and dplyr [30] for data wrangling.

## 3. Results

### 3.1. Study Population

Data from 4056 pregnancies were included in the training dataset (81% of the sample) and 944 in the validation dataset (19% of the sample). In the training dataset, there were 59 (1.5%) PE recorded at any gestational age, 40 (1%) of them delivered at or after 37 weeks (term PE) and 58 (1.4%) of them delivered at or after 34 weeks (late PE) due to PE. In the validation dataset, 23 (2.4%) PE were recorded at any gestational age, including 20 (2.2%) term PE cases.

Complete descriptive demographics for both datasets are available in Table 1. There were several differences in the maternal characteristics of the two datasets, primarily because the center that contributed to the training set specializes in assisted reproductive technologies, resulting in a high proportion of the population coming from that unit. Interestingly, participants in the validation set were generally older, with a higher BMI, more frequently multiparous, more likely to be non-White and to have conceived spontaneously, and showed a higher incidence of chronic hypertension and higher aspirin intake than the training set. Despite these differences, the rate of term PE was lower in the training dataset, likely due to the earlier gestational age at delivery. Ethnicity was described but not used because 98% of the population of both datasets was of White origin.

For longitudinal variables, in the training dataset, only one woman had two measurements, while all others had three or more. In the validation dataset, 13 (1.4%) of the women had two measurements and 931 (98.6%) had three or more.

### 3.2. Joint Model Development

Results of the survival process of the JM are summarized in Table 2. The complete JM is available in Table A1. Significant predictors for developing PE were: non-spontaneous conception (aHR: 2.60; 95% CdI 1.33 to 5.05); history of PE in a previous pregnancy (aHR: 5.98; 95% CdI 1.29 to 22.09); BMI (aHR: 16.60; 95% CdI 5.09 to 52.39); chronic hypertension (aHR: 16.87; 95% CdI 2.13 to 77.44) and its interaction with aspirin (aHR: 0.08; 95% CdI 0.01 to 0.82); first trimester PlGF raw value (aHR: 0.40; 95% CdI 0.21 to 0.77); MAP area (aHR: 1.08; 95% CdI 1.00 to 1.19); and UtA-PI (aHR: 14.08, 95% CdI 2.04 to 103.86).

### 3.3. Predictive Accuracy in the Validation Dataset

The AUROC for all PE delivering at any time after the third-trimester assessment was 0.82 (95% CI 0.72 to 0.92) and the DR was 56.5% (95% CI 34.5 to 76.8) at 10% SPR. The AUROC for term PE was 0.80 (95% CI 0.69 to 0.91), and the DR was 55.0% (95% CI 31.5 to 76.9) at 10% SPR (Table 3).

The calibration plot for all PE predictions is available in Figure 1.

### 3.4. Evolution of AUROC Troughout Pregnancy

As longitudinal measurements are incorporated, the AUROC improves. The values are shown in Figure 2. However, this increase is not linear over time; instead, following a subtle increase in prediction after the 20 weeks’ assessment, a plateau is observed throughout the remaining second trimester, to finally achieve the latest increase from 32 to 36 weeks, likely reflecting time points where the assessments were usually performed.

To summarize, the AUROC values increased from 0.50 (95% CI 0.37 to 0.64) in the first trimester to 0.84 (95% CI 0.73 to 0.94), in the third trimester.

## 4. Discussion

### 4.1. Main Findings

In this study, we have shown that the use of longitudinal measurements of MAP and UtA-PI throughout gestation, based on initial first-trimester baseline measurements, may improve the prediction of late and term PE achieved in the first trimester. During the 35 to 37 weeks assessment, a JM that considers all available information from the first trimester increases the AUROC from 0.50 (95% CI 0.37 to 0.64) to 0.84 (95% CI 0.73 to 0.94).

### 4.2. Comparison with Previous Studies

Screening for PE in the first trimester is widely recommended since aspirin administration to the high-risk group is associated with a 62% reduction in the rate of preterm PE [31]. However, such screening is only able to identify about 40–49% of the term PE [7,11]. We previously examined the performance of first-trimester screening for term PE by three different algorithms [21], including the Fetal Medicine Foundation competing risks model [32], the logistic regression model advocated by Crovetto et al. [8], and the Gaussian distribution one developed by Serra et al. [9]. At 10% fixed SPR, the DRs for term PE for the three approaches were 55.1% (95% CI 48.8 to 61.4); 47.1% (95%CI 40.6 to 53.5); 53.9% (95%CI 47.4 to 60.4), respectively [21], similar to the 55.0% (95%CI 31.5 to 76.9) DR reported in this study. However, there are several differences between those models and the one proposed here. First, except for PlGF in the Crovetto et al. algorithm, all biomarkers in the previously published algorithms were multiples of the median (MoM) transformed, while we used raw values, likely leading to a decreased performance. The reason for this was to examine the possibility of implementing a simpler model that would not require adjustment and strict monitoring. Additionally, we could not include known risk factors like ethnicity, systemic lupus erythematosus (SLE), antiphospholipid syndrome (APS), or family history of PE due to the low number of occurrences in our training sample. However, the same reason makes it unlikely to greatly affect the results. First trimester risk could not be included in the model since not all the necessary variables were available.

Panaitescu et al. assessed the prediction performance of calculating the risk using a combination of maternal factors, MAP, and UtA-PI at the 35–37 weeks’ gestation visit [16]. They obtained a DR of 53.7 (95% CI, 47.6 to 59.7) for a 10% SPR. Similarly, Döbert et al. evaluated a model that used MAP and UtA-PI in the same gestational period, achieving a DR of 59% (95% CI, 48 to 69) and an AUROC of 0.85 (95% CI, 0.80 to 0.89) [18]. In both cases, the results were consistent with our findings.

It is known that models designed to predict term or late PE improve accuracy as gestational age advances and the assessment occurs closer to the event. Longitudinal models allow for the updating of clinical information and changes in the biophysical parameters experienced by the patient to detect women at risk of developing the disease. Previous longitudinal prediction models based on artificial intelligence have shown good performance for late PE [33]. Jhee et al., in a study involving 11,006 pregnancies in Korea where neither PlGF nor sFlt-1 were included like in our case, compared six different algorithms; they used data from the 14th to the 34th week of gestational age, collecting both clinical and biochemical data including routinely measured laboratory data, and achieving an AUROC of 0.924 for late PE for the best model [33].

In contrast, Khalil et al. applied a multilevel mixed-effects linear model in 1069 women and they did not find any improvement by adding the repeated measures of sFlt-1 or PlGF vs. a single measurement, in the prediction of preterm PE [34]. However, they stated that their study was probably underpowered for this purpose.

Finally, Andrietti et al. used the competing risks model in 123,406 women to evaluate the performance of each biomarker (MAP, UtA-PI and PlGF) individually combined with maternal factors, and whether the addition of repeated measures improved the prediction for preterm and term PE, compared to a single measurement [35]. For MAP, the DR of term PE using only one measure at 11–13 weeks was 44.6% (95% CI 39.4 to 49.9) and the DR using three measurements marginally increased the DR to 51.0% (95% CI 45.7 to 56.2), at 10% SPR. For UtA-PI, the DR of term PE using only one measure was 39.9% (95% CI 35.0 to 45.1) and the DR using three measurements was 43.4% (95% CI 38.3 to 48.6), at 10% SPR. For PlGF, the DR of term PE using only one measure was 39.1% (95% CI 30.0 to 49.0) and the DR using three measurements was 53.6% (95% CI 44.0 to 63.0), at 10% SPR.

### 4.3. Strengths and Limitations

The main strength of our study is that it examines a large number of patients receiving several assessments during pregnancy, which included the measurement of several biomarkers according to clinical standards. We also developed and validated the model in two different populations, which ensures the generalizability of the results. Finally, the statistical methods are robust, since they consider the endogenous nature of the longitudinal variables, offering a new perspective that has not been explored before.

The main limitation of the study derives from its retrospective design, which restricts the level of control we have over the data collected, with the immediate consequence of possible non-random data loss. We were unable to estimate sample size or power calculation, but the study’s findings were statistically significant, supporting the strength of the model. The relatively small number of PE cases in our dataset contributes to wide CIs, therefore affecting the precision of our estimates. The performance of the model may have also been underestimated for several reasons. First, we could only use MAP and UtA-PI for longitudinal variables, since PlGF or sFlt-1, which are useful biomarkers in the second and third trimesters, are not routinely measured except for PlGF in the first trimester. Instead, we included UtA-PI, which has limited utility in late gestation [16,18]. Second, to simplify the analysis, we did not apply the MoM transformation to the biomarkers, since our primary goal was to demonstrate the importance of using repeated measurements rather than developing a new model. Last, heterogeneity between the training and validation datasets may have influenced the results with significant differences in baseline characteristics known to be risks for PE; however, this also constitutes a strength, as previously explained. An important limitation to generalization is the lack of racial diversity: with over 98% White participants in both samples, we could not incorporate race as a predictor, limiting the conclusions’ applicability to primarily non-White populations.

### 4.4. Clinical and Research Implications

Continuous assessment of evolving biomarkers allows for reclassification of individual risk when new evidence emerges and subsequent stratification of pregnancy care. This may help clinicians establish a tailored follow-up, adjusted on each visit. Considering not only the new measurements but also the previous ones and the degree of change between assessments, this approach could reassure previously high-risk patients who turn low-risk. Conversely, it would also facilitate closer surveillance of high-risk individuals, who could be instructed on the symptoms or early signs of the disease. This novel approach to managing endogenous variables needs further investigation in prospective clinical trials.

Future research should also consider incorporating additional biomarkers, such as sFlt-1, and PlGF or new biomarkers such as uterine artery volume blood flow in all trimesters, to enhance predictive accuracy [36,37]. However, continuous biomarker assessment in routine clinical settings may face challenges, particularly in low-resource environments or when patient compliance is limited. In such cases, models prioritizing specific combinations of biomarkers could retain the predictive accuracy of classic approaches while accommodating practical constraints. Further research should explore the options for diverse clinical environments.

## 5. Conclusions

Incorporating routinely measured biomarkers throughout pregnancy in a model that accounts for both their actual values and their changes over time may improve the prediction of late and term PE derived from first-trimester assessment. Although we only achieved an AUROC of 0.84 (95% CI 0.73 to 0.94), it was an improvement from the 0.50 (95% CI 0.37 to 0.64) of the first-trimester data. Future research should investigate possible modifications to the proposed model, including new biomarkers, or explore alternative approaches to effectively incorporate repeated measurements of biomarkers.

## Figures and Tables

**Figure 1 medicina-60-01909-f001:**
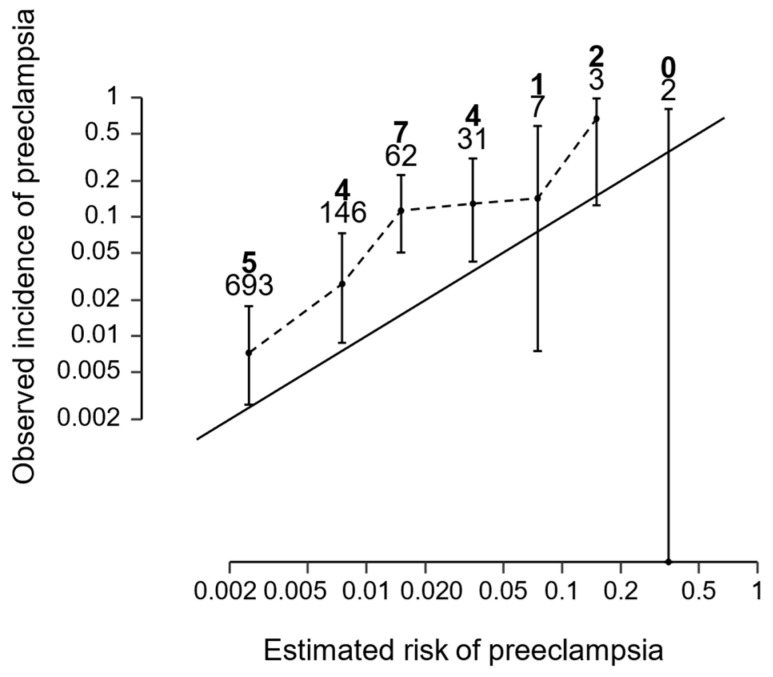
Calibration plot of the models for all preeclampsia delivering after the 35 to 37 weeks assessment. The observed incidence is plotted against the estimated risk. The diagonal black line represents perfect agreement. The numbers in boldface on top indicate the number of preeclampsia cases at each risk bin, and those below are the total number of individuals. Error bars represent 95% confidence intervals. The intercept was 1.24 (95% confidence interval (CI) −0.45 to 2.94) and the slope was 0.98 (95% CI 0.56 to 1.40).

**Figure 2 medicina-60-01909-f002:**
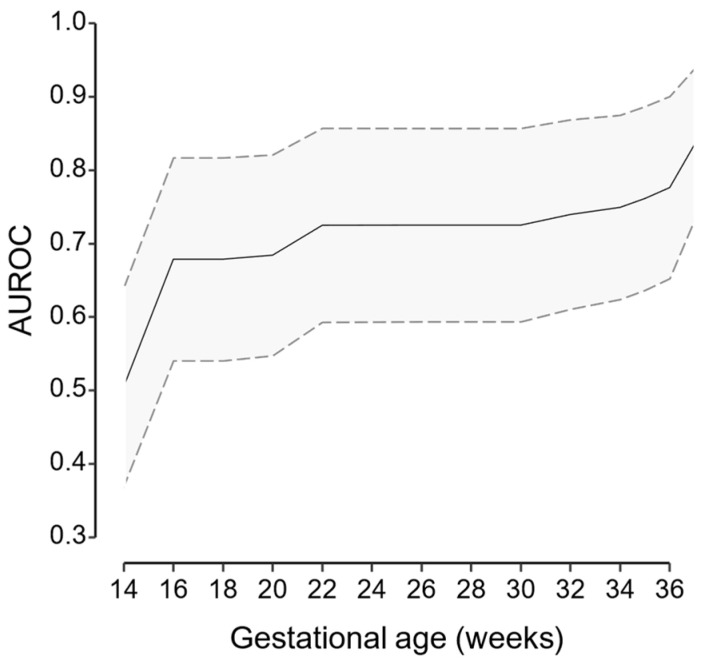
Area under the receiving operator characteristic curve (AUROC), throughout pregnancy. The solid line represents the estimate, and the gray area delimited by dashed lines, the 95% confidence interval.

**Table 1 medicina-60-01909-t001:** Maternal and pregnancy characteristics of the study datasets.

Maternal and Pregnancy Characteristics	Training Set *n* = 4056	Validation Set *n* = 944
Maternal age (years)	31.2 (5.05)	32.7 (5.13) *
Ethnicity		*
White	4050 (99.9%)	933 (98.8%)
Other	6 (0.1%)	11 (1.2%)
Conception		*
Spontaneous	3291 (81.1%)	854 (90.5%)
Assisted	765 (18.9%)	90 (9.5%)
Diabetes mellitus	15 (0.4%)	4 (0.4%)
Chronic hypertension	69 (1.7%)	4 (0.4%) *
Previous preeclampsia		
Nulliparous	2916 (71.9%)	437 (46.3%) *
Multiparous—no previous preeclampsia	1109 (27.3%)	484 (51.3%) *
Multiparous—previous preeclampsia	31 (0.8%)	23 (2.4%) *
Aspirin intake	781 (19.3%)	46 (4.9%) *
Body Mass Index (Kg/m^2^)	22.8 (4.42)	25.5 (5.05) *
Gestational age at delivery (days)	274 (9.98)	277 (9.04) *
Developed preeclampsia	59 (1.5%)	23 (2.4%) *

Mean (SD) or *n* (%). Categorical variables were compared with a two-proportion z-test and continuous variables with the Mann–Whitney test, * indicates *p*-value < 0.05.

**Table 2 medicina-60-01909-t002:** Results of the survival process of the joint model, training sample.

Maternal and Pregnancy Characteristics	Adjusted Hazard Ratio (95% Credible Interval)
Maternal Age (years)	3.17 (0.28 to 37.32)
Conception	
Spontaneous	Reference
Assisted	2.60 (1.33 to 5.05)
Diabetes mellitus	5.85 (0.29 to 42.65)
Chronic hypertension	16.87 (2.13 to 77.44)
Previous preeclampsia	
Multiparous—no previous preeclampsia	Reference
Nulliparous	0.92 (0.47 to 1.85)
Multiparous—previous preeclampsia	5.98 (1.29 to 22.09)
Placental growth factor (UI/mL)	0.40 (0.21 to 0.77)
Body Mass Index (Kg/m^2^)	16.6 (5.09 to 52.39)
Aspirin intake	1.79 (0.91 to 3.51)
Interaction chronic hypertension and aspirin	0.08 (0.01 to 0.82)
Mean arterial pressure (mmHg)	1.08 (1.00 to 1.19)
Uterine Artery Pulsatility Index	14.08 (2.04 to 103.86)

**Table 3 medicina-60-01909-t003:** Performance of the model for the prediction of all preeclampsia (delivering at any gestational age after the 35 to 37 weeks assessment) and term preeclampsia (delivering at ≥37 weeks of gestation).

	All Preeclampsia		Term Preeclampsia	
	Detection Rate (95% CI)	AUROC (95% CI)	Detection Rate (95% CI)	AUROC (95% CI)
10% SPR	56.5 (34.5 to 76.8)	0.84 (0.73 to 0.94)	55.0 (31.5 to 76.9)	0.80 (0.69 to 0.91)
15% SPR	69.6 (471 to 86.8)		65.0 (40.8 to 84.6)	
20% SPR	73.9 (51.6 to 89.8)		70.0 (45.7 to 88.1)	

AUROC: Area Under the Receiving Operator Characteristic curve; CI: Confidence Interval; SPR: Screen-Positive Rate.

## Data Availability

The datasets generated during and/or analyzed during the current study are available from the corresponding authors on reasonable request.

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
