# Peer review of "Continuous Risk Assessment of Late and Term Preeclampsia Throughout Pregnancy: A Retrospective Cohort Study"

_medicina, 2024, doi:10.3390/medicina60121909_

Round 1
Reviewer 1 Report
Comments and Suggestions for Authors
Title: Continuous risk assessment of late and term preeclampsia 2 throughout pregnancy: a retrospective cohort study.
In this article, the authors have tried for diagnostic accuracy of widely available biomarkers with a longitudinal follow up throughout pregnancy to predict all and term preeclampsia. The authors also claims that prediction of term PE in the first trimester is poor, with detection rates is ranging from 40 to 47% with a 10% false positive rate. This necessitates for biomarkers-based research in all and term preeclampsia to reach a clear prediction model through out the time.
My comments are following.
points:
1. What is the main question addressed by the research?
The authors have tried to search or predict for biomarkers-based model starting from the first trimester to full term pregnancy with more sensitive model for preeclampsia. It sounds well.
2. What parts do you consider original or relevant to the field? What
specific gap in the field does the paper address?
It is related to the field of obstetrics and gynae where more sensitive biomarkers based predictive JM model is advised. This contributes to added on novelty. This field is clear and sounds well.
3. What does it add to the subject area compared with other published
material?
This Joint model is based on AUROC which uses the information from the first trimester with 0.5 corelation value where it increased up to average value of 0.84. This model will help predict from the early stage with a peak of 94% value and thus it covers the throughout pregnancy period with good specificity and sensitivity. It gives almost superimposable results with ML models that are often considered black box that make it difficult to interpret the results. Thus, the adopted model sounds well.
4. What specific improvements should the authors consider regarding the
methodology? What further controls should be considered?
Usually, these types of studies are with a prospective cohort model as longitudinal study. The adopted methodology is good and acceptable. Kindly rephrase the comments suggested in the acrobat file of the submitted manuscript to give a clear message in the abstract portion of the manuscript. As per authors claims, it is a retrospective cohort study. Then they shall remove the word longitudinally ……in Abstract section line no 17 page 1, as…..available biomarkers measured through out pregnancy……
5. Are the conclusions consistent with the evidence and arguments presented?
Were all the main questions posed addressed? By which specific experiments?
The conclusion is consistent with model proposed and subsequent results obtained . however, they shall remove the wordings ……Further research should investigate…………..longitudinal data. In lines 331 to 333 of the conclusions section.
These are recommendations.
6. Are the references appropriate?
yes
7. Any additional comments on the tables and figures and the quality of the
data.
Nil

Author Response
Reviewer 1
In this article, the authors have tried for diagnostic accuracy of widely available biomarkers with a longitudinal follow up throughout pregnancy to predict all and term preeclampsia. The authors also claims that prediction of term PE in the first trimester is poor, with detection rates is ranging from 40 to 47% with a 10% false positive rate. This necessitates for biomarkers-based research in all and term preeclampsia to reach a clear prediction model through out the time.
My comments are following.
Comment 1: 1. What is the main question addressed by the research?
The authors have tried to search or predict for biomarkers-based model starting from the first trimester to full term pregnancy with more sensitive model for preeclampsia. It sounds well.
Response 1: Thank you
Comment 2: 2. What parts do you consider original or relevant to the field? What specific gap in the field does the paper address?
It is related to the field of obstetrics and gynae where more sensitive biomarkers based predictive JM model is advised. This contributes to added on novelty. This field is clear and sounds well.
Response 2: Thank you
Comment 3: 3. What does it add to the subject area compared with other published material?
This Joint model is based on AUROC which uses the information from the first trimester with 0.5 corelation value where it increased up to average value of 0.84. This model will help predict from the early stage with a peak of 94% value and thus it covers the throughout pregnancy period with good specificity and sensitivity. It gives almost superimposable results with ML models that are often considered black box that make it difficult to interpret the results. Thus, the adopted model sounds well.
Response 3: Thank you
Comment 4: 4. What specific improvements should the authors consider regarding the methodology? What further controls should be considered?
Usually, these types of studies are with a prospective cohort model as longitudinal study. The adopted methodology is good and acceptable. Kindly rephrase the comments suggested in the acrobat file of the submitted manuscript to give a clear message in the abstract portion of the manuscript. As per authors claims, it is a retrospective cohort study. Then they shall remove the word longitudinally ……in Abstract section line no 17 page 1, as…..available biomarkers measured through out pregnancy……
Response 4: Thank you for the insight, we have rephrased the sentence to clarify it. This is a longitudinal retrospective cohort study, to avoid confusion with a transversal retrospective cohort study.
Comment 5: 5. Are the conclusions consistent with the evidence and arguments presented? Were all the main questions posed addressed? By which specific experiments?
The conclusion is consistent with model proposed and subsequent results obtained . however, they shall remove the wordings ……Further research should investigate…………..longitudinal data. In lines 331 to 333 of the conclusions section.
These are recommendations.
Response 5: Thank you, we have replaced longitudinal data by “repeated measurements” to avoid confusion
Comment 6: 6. Are the references appropriate?
Yes
Response 6: Thank you
Comment 7: 7. Any additional comments on the tables and figures and the quality of the
data.
Nil
Response 7: We are very grateful to the reviewer for their nice comments
Reviewer 2 Report
Comments and Suggestions for Authors
Dear Authors,
Thank you for your submission, which represents a very important investigation into the continuous risk assessment of the prediction of preeclampsia during pregnancy by use of longitudinal biomarkers' measurements. In fact, it is an essential topic if we take into account implications relative to maternal-fetal health. I have several comments related to the Major Revision in the interest of clarity, transparency, and robustness.
1. Materials and Methods
Surveillance Population and Data Retrieval: The eligibility criteria of patients are not concisely stated. Implications for generalization require an explanation because of the low percentage of non-White participants.
2. Biomarker Measurement and Selection Rationale Most importantly, continuing, there is the need to give a rationale for why certain biomarkers were chosen, such as MAP, UtA-PI, and not others such as PlGF and sFlt-1 in later trimesters that could have theoretically altered the predictive accuracy of these models. If appropriate, discuss whether in future studies sFlt-1 and PlGF could theoretically be integrated into future analyses to even further enhance results​.
3. Sample Size and Power Analysis: The sample size calculation must be clearly described along with the discussion on the statistical power achieved. This would help the reader better understand the strength and limitations of this study, in particular, considering the limited incidence of PE in the validation set​.
4. Ethical Considerations: Though the article mentioned ethical approval, it would be much clearer if it gave more details on informed consent, respect for persons in regard to participation or not in this study. Without a doubt, the protocol number has to be indicated.
5. Resultados
Descriptive Statistics: Provide an extended table of demographic baseline and clinical characteristics. Highlight large differences in training and validation datasets, which will allow for investigation of possible biases according to retrospective design and dataset origin​.
6. AUROC and Predictive Value Analysis: All metrics, including AUROC values and detection rates, should be represented with confidence intervals in a uniform manner. It would also be great if it were explained clearly how exactly the AUROC improved over time within the different trimesters-what kind of increase, maybe-elaborating in detail the insights related to predictive improvements across each trimester.
7. Clinical Relevance and Applicability: Sections related to the operability of the section on continuous biomarker assessment in a routine clinical setting when resources are limited or when there is a lack of compliance on the part of the patient would be more usefully elaborated. This would enhance the discussion of the practical relevance of the model​.
Limitations and Future Directions: Please give more detail on these limitations in view of the retrospective nature, heterogeneous datasets, and inability to include such factors like ethnicity due to limited diversity. Cite novel article about the uterine blood flow and pregnancy. The uterine blood volume could be a future topic and it's citation could help additional study about PE and uterine blood flow volume:
- https://doi.org/10.1016/j.placenta.2011.04.004
- DOI: 10.1055/a-2075-3021
I am sure that putting the above recommendations into practice will greatly enhance the clarity, impact, and applicability of the findings.
Author Response
Reviewer 2
Thank you for your submission, which represents a very important investigation into the continuous risk assessment of the prediction of preeclampsia during pregnancy by use of longitudinal biomarkers' measurements. In fact, it is an essential topic if we take into account implications relative to maternal-fetal health. I have several comments related to the Major Revision in the interest of clarity, transparency, and robustness.
Comment 1: Materials and Methods
Surveillance Population and Data Retrieval: The eligibility criteria of patients are not concisely stated. Implications for generalization require an explanation because of the low percentage of non-White participants.
Response 1: Thank you. The eligibility criteria are specified in the first paragraph of methods, but we have slightly modified them to make them clearer: “All pregnant women ≥ 18 years old, with singleton pregnancies and non-malformed live fetuses were invited to participate in a larger study for the prediction and prevention of pregnancy complications. For this study, we included only women undergoing first-trimester assessment of PE, known perinatal outcome and at least one more ultra-sound assessment performed during the second and/or third trimesters of their pregnancy, where MAP and UtA-PI had been evaluated.”
We have added the following sentence regarding the implications of the low percentage of non-white participants in the discussion:
“An important limitation to generalization is the lack of racial diversity: with over 98% White participants in both samples, we could not incorporate race as a predictor, limiting the conclusions’ applicability to primarily non-White populations.”
Comment 2: Biomarker Measurement and Selection Rationale Most importantly, continuing, there is the need to give a rationale for why certain biomarkers were chosen, such as MAP, UtA-PI, and not others such as PlGF and sFlt-1 in later trimesters that could have theoretically altered the predictive accuracy of these models. If appropriate, discuss whether in future studies sFlt-1 and PlGF could theoretically be integrated into future analyses to even further enhance results​.
Response 2: Thank you for your comment. We have clarified in the methods that we chose the biomarkers solely based on availability:
“We evaluated all biomarkers available in each assessment which included maternal characteristics and history, PlGF, MAP, UtA-PI obtained during the first trimester, and maternal weight, MAP and UtA-PI measured in the second and third trimesters.”
We had also mentioned in the limitations that “we could only use MAP and UtA-PI for longitudinal variables, since PlGF or sFlt-1, which are useful biomarkers in the second and third trimesters, are not routinely measured except for PlGF in the first trimester. Instead, we included UtA-PI which has limited utility in late gestation[16],[18].”
Finally, we have added the following sentence at the end of “Clinical and research implications”: “Future research should also consider incorporating additional biomarkers, such as sFlt-1, PlGF or new biomarkers such as uterine artery volume blood flow in all trimesters, to enhance predictive accuracy [36, 37].”
Comment 3: Sample Size and Power Analysis: The sample size calculation must be clearly described along with the discussion on the statistical power achieved. This would help the reader better understand the strength and limitations of this study, in particular, considering the limited incidence of PE in the validation set​.
Response 3: Thank you, we have added the following sentences in the methods and discussion:
“We opted to use all available data since standard calculations for sample size and power are not directly applicable to JM due to the dependency of the observations.”
“We were unable to estimate sample size or power calculation, but the study’s findings were statistically significant, supporting the strength of the model.”
Comment 4: Ethical Considerations: Though the article mentioned ethical approval, it would be much clearer if it gave more details on informed consent, respect for persons in regard to participation or not in this study. Without a doubt, the protocol number has to be indicated.
Response 4: Thank you, we have added an Ethics Committee Statement, including the name of the committee, the approval code and the date: “Ethics Committee Statement: The study was conducted in accordance with the Declaration of Helsinki, and approved by the Ethics Committee of Hospital Universitario Torrevieja and Hospital Universitario Elche-Vinalopó (protocol code 2019.059-OC and date of approval October 25, 2019).”
Comment 5: Descriptive Statistics: Provide an extended table of demographic baseline and clinical characteristics. Highlight large differences in training and validation datasets, which will allow for investigation of possible biases according to retrospective design and dataset origin​.
Response 5: Thank you. Table 1 shows a description of maternal and pregnancy characteristics according to set. We do not have a lot more variables available but if there is any of particular interest, we will do our best to try to incorporate it to the table.
For the differences, we have added a sentence in the text highlighting the most notable ones between the training and validation sets:
“Participants in the validation set were generally older, with a higher BMI, more frequently multiparous, more likely to be non-White and to have conceived spontaneously, showed a higher incidence of chronic hypertension and higher aspirin intake than the training set.”
Comment 6: AUROC and Predictive Value Analysis: All metrics, including AUROC values and detection rates, should be represented with confidence intervals in a uniform manner. It would also be great if it were explained clearly how exactly the AUROC improved over time within the different trimesters-what kind of increase, maybe-elaborating in detail the insights related to predictive improvements across each trimester.
Response 6: Thank you for noticing, we have reviewed the manuscript to ensure that both, confidence and credible intervals are consistent.
We have expanded the current explanation for the increase in the AUROC: “However, this increase is not linear over time, instead, following a subtle increase in prediction after the 20 weeks’ assessment, a plateau is observed throughout the remaining second trimester, to finally achieve the latest increase from 32 to 36 weeks, likely reflecting time points where the assessments were usually performed.”
Comment 7: Clinical Relevance and Applicability: Sections related to the operability of the section on continuous biomarker assessment in a routine clinical setting when resources are limited or when there is a lack of compliance on the part of the patient would be more usefully elaborated. This would enhance the discussion of the practical relevance of the model​.
Response 7: Thank you, we have added a paragraph about this in “Clinical and research implications”
“Future research should also consider incorporating additional biomarkers, such as sFlt-1 and PlGF in later trimesters, to enhance predictive accuracy, or even evaluate new biomarkers such as uterine artery volume blood flow [36,37]. However, continuous biomarker assessment in routine clinical settings may face challenges, particularly in low-resource environments or when patient compliance is limited. In such cases, models that prioritize specific combinations of biomarkers could maintain predictive accuracy while accommodating practical constraints. Further research should explore the options for diverse clinical environments.”
Comment 8: Limitations and Future Directions: Please give more detail on these limitations in view of the retrospective nature, heterogeneous datasets, and inability to include such factors like ethnicity due to limited diversity. Cite novel article about the uterine blood flow and pregnancy. The uterine blood volume could be a future topic and it's citation could help additional study about PE and uterine blood flow volume:
- https://doi.org/10.1016/j.placenta.2011.04.004
- DOI: 10.1055/a-2075-3021
Response 8: Thank you, we have expanded the limitations to include more details and edited the “Clinical and research implications” to include the uterine artery volume blood flow.
Limitations:
“The main limitation of the study derives from its retrospective design, which restricts the level of control we have over the data collected, with the immediate consequence of possible non-random data loss. We were unable to estimate sample size or power calculation, but the study’s findings were statistically significant, supporting the strength of the model. The relatively small number of PE cases in our dataset contributes to wide CIs, therefore affecting the precision of our estimates. The performance of the model may have also been underestimated for several reasons. First, we could only use MAP and UtA-PI for longitudinal variables, since PlGF or sFlt-1, which are useful biomarkers in the second and third trimesters, are not routinely measured except for PlGF in the first trimester. In-stead, we included UtA-PI which has limited utility in late gestation [16],[18]. Second, to simplify the analysis, we did not apply the MoM transformation to the biomarkers, since our primary goal was to demonstrate the importance of using repeated measurements rather than developing a new model. Last, heterogeneity between the training and validation datasets may have influenced the results with significant differences in baseline characteristics known to be risks for PE; however, this also constitutes a strength as previously explained. An important limitation to generalization is the lack of racial diversity: with over 98% White participants in both samples, we could not incorporate race as a predictor, limiting the conclusions’ applicability to primarily non-White populations.”
Please, see comment above for the clinical and research implications paragraph.
Comment 9: I am sure that putting the above recommendations into practice will greatly enhance the clarity, impact, and applicability of the findings.
Response 9: Thank you very much for your comments. Indeed, we believe they helped to substantially improve the manuscript.
Round 2
Reviewer 2 Report
Comments and Suggestions for Authors
Dear Authors,
I appreciate the extensive revisions and the attempt to address comments given in the previous review of your manuscript, "Continuous risk assessment of late and term preeclampsia throughout pregnancy-a retrospective cohort study."
After re-reviewing the revised submission carefully, I am gratified to observe that most comments have been attended to satisfactorily. One will find in the revised manuscript the following:
More detail on patient eligibility and more descriptive definitions are required to characterize the study population.
Informed consent was obtained from the participants, and all ethical considerations were described appropriately along with the approval of the ethics committee, stating the statement of the protocol number.
The expanded tables and statistics that give an overview of the demographic nature and most important differences in the training and validation datasets.
The inclusion of confidence intervals of the AUROC metrics and further explanations in terms of the improvement in predictive accuracy across trimesters. You also give a richer discussion of the clinical applicability of your results, which gives greater practical relevance to the study.
Congratulations on such vital work; I look forward to the impact it would have in the field. Best regards,